# A Sensitive Liquid Chromatography–Tandem Mass Spectrometry Method for Measuring Fosfomycin Concentrations in Human Prostatic Tissue

**DOI:** 10.3390/pharmaceutics16050681

**Published:** 2024-05-17

**Authors:** Matteo Conti, Beatrice Giorgi, Rossella Barone, Milo Gatti, Pier Giorgio Cojutti, Federico Pea

**Affiliations:** 1Clinical Pharmacology Unit, IRCCS Azienda Ospedaliero-Universitaria di Bologna, 40138 Bologna, Italy; beatrice.giorgi@aosp.bo.it (B.G.); rossella.barone3@unibo.it (R.B.); milo.gatti2@unibo.it (M.G.); piergiorgio.cojutti@unibo.it (P.G.C.); federico.pea@unibo.it (F.P.); 2Department of Medical and Surgical Sciences, Alma Mater Studiorum, University of Bologna, 40138 Bologna, Italy

**Keywords:** fosfomycin, therapeutic drug monitoring, micro samples, prostatic tissue, liquid chromatography-tandem mass spectrometry

## Abstract

The aim of this study was to develop and validate a fast and sensitive bioanalytical method for the accurate quantification of fosfomycin concentrations in human prostatic tissue. The sample preparation method only required milligrams of tissue sample. Each sample was mixed with two times its weight of water and homogenized. A methanol solution that was three times the volume of the internal standard (fosfomycin-13C3) was added, followed by vortex mixing and centrifugation. After its extraction from the homogenized prostatic tissue, fosfomycin was quantified by means of a liquid chromatography–tandem mass spectrometry (LC-MS/MS) triple quadrupole system operating in negative electrospray ionization and multiple reaction monitoring detection mode. The analytical procedure was successfully validated in terms of specificity, sensitivity, linearity, precision, accuracy, matrix effect, extraction recovery, limit of quantification, and stability, according to EMA guidelines. The validation results, relative to three QC levels, were 9.9% for both the within-day and inter-day accuracy (BIAS%); 9.8% for within-day precision; and 9.9 for between-day precision. A marked matrix effect was observed in the measurements but was corrected by normalization with the internal standard. The average total recovery was high (approximatively 97% at the three control levels). The dynamic range of the method was 0.1–20 μg/g (R^2^ of 0.999). Negligible carry-over was observed after the injection of highly concentrated samples. F in the sample homogenate extracts was stable at 10 °C and 4 °C for at least 24 h. In the tissue sample freeze–thaw experiments, a significant decrease in F concentrations was observed after only two cycles from −80 °C to room temperature. The novel method was successfully applied to measure fosfomycin in prostatic tissue samples collected from 105 patients undergoing prostatectomy.

## 1. Introduction

Pre-operative antibiotic prophylaxis is a well-recognized and widely used strategy in urology for preventing infectious complications resulting from diagnostic or therapeutic procedures. The emergence of multidrug-resistant (MDR), Gram-negative pathogens causing healthcare-associated urinary tract infections is, worryingly, increasing nowadays, and is making the selection of an appropriate antibiotic [1,2,3] challenging. Traditional antimicrobials, like cotrimoxazole, second-generation cephalosporins, and fluoroquinolones, are only effective against wild-type Gram-negatives and should no longer be recommended in this setting [4].

In this scenario, fosfomycin may represent a useful option as it shows a wide spectrum of activity against MDR Gram-negative uropathogens, including ESBL producers and even carbapenemase producers [5,6,7]. Fosfomycin trometamol (F-t) is an old oral formulation of fosfomycin that, after intestinal absorption, is rapidly converted to the fosfomycin active moiety. Several clinical studies showed that prophylactic use of F-t, before procedures like transurethral resection (TURP) and/or holmium laser enucleation of the prostate (HoLEP), reduced the incidence of both symptomatic UTIs and asymptomatic bacteriuria in the first post-intervention week [8,9,10].

It was shown that fosfomycin has a good penetration rate into the prostatic tissue [11,12]. Considering that fosfomycin plasma concentrations may represent only an unreliable surrogate of tissue exposure, determining absolute fosfomycin concentrations in prostatic tissue is strictly required for assessing optimal PK/PD targets at suspected and/or documented sites of infection. For better defining the prostatic tissue profile of fosfomycin, measuring fosfomycin concentrations in the prostatic tissue by collecting bioptic samples with a sparse strategy in a large number of patients undergoing TURP or HoLEP may be a valuable approach. This would allow for the development of a population pharmacokinetic/pharmacodynamic (PK/PD) model for predicting the probability of the target attainment of effective fosfomycin exposure in prostatic tissue after oral F-t administration. For this purpose, a bioanalytical assay that enables direct and accurate quantification of fosfomycin concentrations in bioptic samples is needed.

Various analytical methods to determine F concentrations in biological samples are currently available and have been recently summarized [13]. With regard to the specific determination of fosfomycin in prostatic tissue, only a few reports have been published in the literature thus far. Specifically, one recent paper indirectly estimated fosfomycin concentrations in prostatic tissue [14]. In another study, an LC-MS/MS assay was applied to measure F concentrations in rat biopsies [15]. In two other studies, fosfomycin concentrations were assessed in human prostatic tissue with the same analytical technique [13]. By all accounts, the LC-MS/MS has clearly emerged as the best performing analytical technique with which to determine fosfomycin concentrations in biological samples [13]. Furthermore, a very sensitive LC-MS/MS method for the determination of F in human plasma microsamples was recently developed and validated by applying an innovative reverse-phase chromatographic approach [16]. By applying this latter approach, the aim of this study was therefore to develop and validate an efficient bioanalytical assay to measure fosfomycin concentrations in human prostatic tissue.

## 2. Materials and Methods

### 2.1. Chemical and Reagents

Fosfomycin (Mw 138.06 g/mol) tromethamine salt (F-t) and [13C3]-rac Fosfomycin (Mw 141.04 g/mol) (FIS) (Figure 1) were purchased from Alsachim (Illkirch, France). Liquid chromatography–MS/MS-grade water (ultra-pure water) was produced by a Milli-Q^®^ Direct system (Millipore Merck—Darmstadt, Germany). All other reagents were purchased from CHROMASOLV™ (Thermofisher Scientific, Milan, Italy).

### 2.2. Tissue Samples

Prostatic tissue samples were collected from patients receiving F-t prophylaxis and subsequently undergoing elective TURP or HoLEP procedures for diagnostic/therapeutic purposes. For blanking purposes, prostatic tissue bioptic samples collected from patients receiving other types of antimicrobial prophylaxis were used. Samples were immediately stored at −80 °C and subsequently processed.

### 2.3. Sample Treatment

After defrosting, each prostatic tissue sample was weighted, put into a 12 × 75 mm glass vial, and mixed with ultrapure milliQ^®^ water in a 1:2 weight proportion. The mixture was ground in an IKA^®^ T10 basic ULTRA-TURRAX^®^ laboratory homogenizer for 3 min.

Three µL of the tissue homogenate were diluted with 47 µL of ultrapure milliQ^®^ water to yield a final volume of 50 µL. A total of 150 μL of the FIS-methanol solution was added to induce protein precipitation and fosfomycin extraction. The solution was vortexed for 20 s and then centrifuged for 5 min at 10,800× *g* at room temperature. Subsequently, 100 µL of the clear supernatant was transferred into an autosampler vial and 3 µL was injected into the LC-MS/MS system. Final fosfomycin concentrations were normalized per gram of prostatic tissue.

### 2.4. Stock Solutions, Standards, and Quality Controls

Eight fosfomycin stock solutions were prepared in MilliQ water at the following concentrations: 10,000 mg/L (A); 5000 mg/L (B); 2500 mg/L (C); 500 mg/L (D); 250 mg/L (E); 100 mg/L (F); 10 mg/L (G); and 1 (H) mg/L. Calibrators were obtained by spiking F-free tissue homogenates with the F stock solutions, and subsequently stored at room temperature for at least 30 min in order to ensure equilibration of the drug in the blank tissue homogenate. The calibration was performed by using 6 calibrators (calibration points: 0.1–0.5–1–5–10–20 µg/g), ranging from 0.1 to 20 µg/g.

Three quality-control (QC) samples were prepared at 0.15 µg/g (Low QC, LQC), 1.5 µg/g (Medium QC, MQC) and 15 µg/g (High QC, HQC) with different stock solutions and blank tissue samples.

Single-use aliquots of the stock solutions, calibrators, and controls were stored at −80 °C for subsequent use.

### 2.5. Instrumentation

Measurements were performed by an LC-MS/MS system comprising an Agilent 6495 triple quadrupole mass spectrometer coupled to an Agilent 1290 ultra-high performance liquid chromatography instrument.

Tandem-mass-spectrometry detection and acquisition was performed in multiple reaction monitoring (MRM) mode and negative electrospray ionization (ESI). Jet-spray parameters were set as follows: gas temp = 200 °C, gas flow = 14 L/min, nebulizer pressure = 35 psi, sheath gas temp = 300 °C, sheath gas flow = 11 L/min, capillary voltage: positive = 4000 V, Nozzle voltage = 0 V, negative = −3000 V, Nozzle voltage = 1500 V. The fragmentor voltage is a fixed (compound-insensitive) parameter that is determined during the mass calibration procedure. The precursor and product ion masses, and the collision energy, are compound-sensitive and were therefore optimized.

MRM transition specific parameters are reported in Table 1.

Chromatography was performed by means of an Agilent 1295 U-HPLC equipped with a ZORBAX Eclipse plus C18 column (2.1 × 50 mm, 1.8 µm particle size; Agilent, Santa Clara, CA, USA) kept at 25 °C. Analyte separation was performed at a constant flow rate of 0.5 mL/min, in binary gradient elution mode (Table 2) with mobile phases A (composed of water-formic acid 100:0.1, *v*/*v*) and B (methanol-formic acid 100:0.1, *v*/*v*). Sample extracts were injected by the autosampler where samples were maintained at 10 °C.

Chromatographic data acquisition, peak integration, and quantification were performed by means of MassHunter software 10.0 (Agilent, Santa Clara, CA, USA). For illustrative purposes, the MRM ion extraction chromatograms reported in Figure 1 were prepared using Graphpad Prism software 10.2.0 (GraphPad Software, Boston, MA, USA).

### 2.6. Method Validation

Validation was performed according to the European Medicines Agency (EMA) guidelines for bioanalytical methods. Selectivity, linearity, accuracy, precision, lower limit of quantification (LLOQ), recovery, matrix effect, and stability were calculated [17].

#### 2.6.1. Sensitivity

Sensitivity was set equal to the lower calibration level in the dynamic range, provided that the signal to noise (S/N) of the analyte was >10. The S/N was calculated using MassHunter software through an automatic command after selecting the analyte peak and two intervals ten seconds before and after the analyte peak itself.

#### 2.6.2. Selectivity and Carry-Over

Analysis was performed on ten different blank tissue samples to check for potentially interfering signals, at the retention times of both the fosfomycin and its internal standard, due to the prostatic tissue matrix components. Carry-over was assessed by injecting blank sample extracts after the upper limit of quantification (ULOQ) and was considered acceptable if the detected signal was <20% of that of the lower limit of quantification (LLOQ).

#### 2.6.3. Linearity and Limit of Quantification (LOQ)

Six calibrators were generated by spiking blank matrices with F ranging from 0.1 to 20 µg/g, which were used to calibrate the system, as shown in the example reported in Figure 2. Linearity of the calibration curve was evaluated according to the fitness-for-purpose approach [18]. The LLOQ was set as the concentration of the lowest calibrator, provided that its signal to noise was >10.

#### 2.6.4. Dilution Integrity

Dilution integrity was verified in terms of accuracy and/or precision in order to allow for the dilution of samples having concentrations above the ULOQ. Dilution was performed with a drug-free matrix homogenate prepared according to the method described for sample preparation. For this purpose, three different blank samples spiked at 80 µg/g were analyzed in triplicate (n = 9). The limits of acceptability were CV < 15% for precision and ±15% of the nominal concentration for accuracy.

#### 2.6.5. Precision and Accuracy

Precision (mean CV%) and accuracy (mean BIAS%) were assessed at three control levels: the LQC, the MQC, and the HQC, five times in a single day (intra-day), and over three different days (inter-day).

#### 2.6.6. Matrix Effect and Extraction Recovery

The percent matrix effect (ME) and the extraction recovery (ER) were assessed at three control levels: LQC, the MQC and the HQC by means of the following equations:ME (%) = C/A × 100;ER (%) = B/C × 100.

IS normalized percent matrix effect (ISn-ME) and extraction recovery (ISn-ER) were assessed by means of the following equations:ISn-ME (%) = C1/A1 × 100;ISn-ER (%) = B1/C1 × 100.

where:

A = Fosfomycin peak area obtained by injecting water-methanol 1:3 *v*/*v* samples (n = 3) spiked at the three QC levels.

A1 = Fosfomycin peak area/IS peak area obtained by injecting water-methanol 1:3 *v*/*v* samples (n = 3) spiked at the three QC levels.

B = Fosfomycin peak area obtained in drug-free tissue extracts (n = 3) spiked at the three QC levels before extraction.

B1 = Fosfomycin peak area/IS peak area obtained in drug-free tissue extracts (n = 3) spiked at the three QC levels before extraction.

C = Fosfomycin peak area obtained by a drug-free tissue extract (n = 3) spiked at the three QC levels after the extraction.

C1 = Fosfomycin peak area/IS peak area obtained in drug-free tissue extracts (n = 3) spiked at the three QC levels after the extraction.

These measurements were performed in triplicate on ten F blank samples (n = 30) to take into account the matrix composition variability.

#### 2.6.7. Stability

The stability of fosfomycin at the LQC, the MQC and the HQC was assessed in extracts and in samples in the following storage conditions:extracts kept on board at 10 °C for 24 h;extracts kept in a freezer at −20 °C for 24 h;samples (homogenized) undergoing three complete freeze and thaw cycles from −80 °C to 25 °C.

The concentrations detected in these extracts and samples were compared with the nominal ones to assess the stability of the analyte.

## 3. Results

### 3.1. Optimization of LC-MS/MS Conditions

Detection of the analyte and the internal standard was performed by single charge negative ion mass transitions. F and IS were detected by means of the 137.0 > 79.0 and 140.0 > 79.0 *m*/*z* transitions, respectively. These were determined by inspecting the MS/MS fragmentation-pattern spectra of the analytes and data available in the literature [17,18,19,20,21,22]. The parameters reported in Table 2 are those determining optimal sensitivity (higher signals) and specificity (absence of interferences).

The binary gradient elution, according to the program reported in Table 1, creates sharp peaks in a short chromatographic run time (i.e., 2.5 min). Reproducibility of the retention times was verified throughout the analytical runs. This reproducibility was obtained by applying the reconditioning step indicated in Table 1, which consisted of 0.5 min at 0.5 mL/min flow of 5% of the B phase to allow for proper column reconditioning between runs.

MRM transitions were highly specific. Fosfomycin-free MRM chromatograms showed the presence of an IS peak at 0.7 min. The presence of an isobaric peak at 0.4 min (Figure 1A) both in blanks, in the calibrators, the controls and the patient’s samples was always detected (Figure 1A–C). This peak must therefore be attributed to an unrecognized molecular component of the prostatic tissue. However, the accuracy of fosfomycin detection was not influenced by it, since it is neatly separated from the peak at 0.7 min typical of F and FIS.

MRM chromatograms of LLOQ samples (Figure 1B) showed high S/N ratio (88.5) for the fosfomycin peak measured by the Mass Hunter software routine by selecting the reference signal ten seconds before and after the peak. Patient’s sample MRM chromatograms (Figure 1C) showed optimal peak shape and resolution for F and the IS.

### 3.2. Method Validation

#### 3.2.1. Sensitivity and LOQ

The LLOQ was set equal to the lowest calibration point, i.e., 0.1 µg/g. The S/N ratio for this concentration level was equal to 88.5 (Figure 1B). This S/N indicates that analytical sensitivity could be set at a much lower concentration level. However, validation of a better LLOQ was not pursued in this study due to the fact that the adopted LLOQ (0.1 μg/g) was sufficient for sample evaluation and lower fosfomycin concentrations are not significant for our clinical purposes.

#### 3.2.2. Selectivity and Carry-Over

The absence of interfering peaks at the retention time range of the analyte, in the MRM chromatograms of blank prostate tissue samples, (as in the example shown in Figure 1A) support the specificity of the operative conditions adopted.

The carry-over effect was acceptable since ten MRM chromatograms of the fosfomycin-free sample recorded after running a ULOQ sample showed peak-area values for F equal to 9.5% on average in the area of the ten LLOQ peak areas.

#### 3.2.3. Linearity and Dilution Integrity

The calibration curve was built by plotting the F peak area/IS peak area over the corresponding nominal concentrations. Linearity of the fitting was excellent (with R^2^ = 0.99), as shown in Figure 2. Fitting was performed by applying a 1/x weighting, the default option in the software. The equation y = 36.274 x + 0.007 was calculated by pooling the data obtained in seven different days and the fitness-for-purpose approach verified the acceptability of the linear model used for the calibration.

Dilution integrity was verified by calculating the average accuracy and precision in the measurement of three independent samples spiked at 80 µg/g of fosfomycin. The mean BIAS was equal to −5.1% and mean CV% was equal to 7.5%.

#### 3.2.4. Accuracy and Precision

Precision (mean CV%) and accuracy (mean BIAS%), obtained at the three QC levels, are reported in Table 3. The intra- and inter-day coefficients of variation of the different QC levels ranged from 2.8% to 8.6%, and from 4.1% to 9.9%, respectively. The intra- and inter-day accuracy biases of the different QC levels ranged from 8.2% to 9.9%, and from to 1.7% to 8.8%, respectively. All parameters therefore fulfilled the EMA requirements for successful validation.

#### 3.2.5. Matrix Effect and Extraction Recovery

Percent ME and ER were assessed at the three control levels (LQC, MQC and the HQC) (Table 4). ME was relevant, the ISn-ME (after normalization with the internal standard) values met the EMA criteria for validation.

The ER was high at all three of the levels tested (ranging from 96.8% to 97.8%), meeting the EMA criteria for validation, and became even higher after normalizing with the internal standard (ISn-ER). The variations at different concentrations were negligible.

#### 3.2.6. Stability

F stability was tested in triplicate (n = 3) at the three QC levels in different operating conditions, as reported in Table 5. A consistent decrease in F concentrations (below a 10% threshold of the nominal value) was observed after two freeze–thawing cycles from −80 °C to room temperature.

Sample extracts were stable for at least 12 h when kept onboard at 10 °C inside the autosampler plate, allowing for reliable sample injection and re-injection within a useful timeframe. F stability was ensured up to 24 h when these extracts were stored in a refrigerator at −20 °C, in case the analysis needed to be performed on the day after sample preparation.

#### 3.2.7. Concentration of Fosfomycin in Prostatic Tissue

The box-and-whisker plot of the fosfomycin concentrations measured in the prostatic tissue of the patients included in the current study is shown in Figure 3. The values range from a minimum of 1.2 µg/g to a maximum of 78.3 µg/g, with a median of 14.7 µg/g and a 25th–75th interquartile range of 5.4–17.95 µg/g.

## 4. Discussion

We developed and validated an accurate, precise, and sensitive bioanalytical method for measuring F concentrations in human prostatic tissue. To the best of our knowledge, this is the first LC-MS/MS method that has been fully validated for these specific tasks, although other methods were previously employed [15,17].

In this study, unlike a study previously performed by Gardiner et al. [18], we did not differentiate between histological zones (i.e., the transition and the peripheral zone) of the prostatic tissue as such a discrimination was outside the scope of this study.

Sample preparation was straightforward, as the tissue sample was mixed with water (2:1 in weight) before mechanical grinding; the homogenate was further diluted with water (50:1 in weight), mixed on a vortex before the final protein crash-down/extraction step performed with a methanol solution of the FIS.

For the analysis, a reverse-phase chromatographic approach similar to one previously employed by us for measuring F concentrations in plasma was adopted [16]. Chromatography was performed in a simple, binary, gradient elution mode, with water and methanol as the mobile phases, and a standard 50 × 2.1 mm reverse-phase C18 column as the stationary phase. This straightforward chromatographic approach is different than the approach employed by other authors that is based on HILIC chromatography. This latter type of chromatography could theoretically be more appropriate, according to the hydrophilic nature of the F molecule [19]. However, our findings suggest that the reverse-phase approach may be as effective as the HILIC approach when combined with the fast binary gradient used in this study. In addition, the reverse-phase approach may be useful in laboratory practices, as it allows for the same chromatographic column to be used between runs when dealing with the analysis of different multiple drugs. It is also noteworthy that the chromatographic run was as short as 2.5 min, which is much faster than those reported for other LC-MS/MS methods [19,20,21]. This methodological feature may also be particularly helpful in clinical practices, as it speeds up the turn-around times of TDM results.

The analytical specificity of the method was assessed by analyzing various prostatic samples of patients undergoing other antibiotic regimens (F blank samples). The absence of any interfering peak in the presence of various endogenous compounds and/or administered drugs demonstrated that the method was, indeed, highly specific.

However, with regard to specificity, it should be noted that an isobaric peak was retrieved at the retention time that was equal to 0.4 min in the FIS-related MRM chromatograms of all (blanks, samples, calibrators, and controls) prostatic extracts. This peak is obviously associated with an isobaric interference specifically present in the prostatic extracts. However, as this peak was neatly separated from those recorded at 0.7 min for F and FIS, it did not interfere in either the analysis or the validation process.

A marked matrix effect, which was expected due to the presence of the variety of molecular components in the crude tissue extracts, was clearly observed, as has been reported for other biological matrices as well [19,21,22]. However, the use of an isotopically labeled analogue of F was very effective in completely compensating the matrix effect.

A high extraction-recovery yield (>90%) was calculated at all tested concentrations. This high extraction yield, obtained without a dedicated extraction step for the analyte, may be due to the highly hydrophilic nature of the F molecule and the composition of the extraction mixture employed by us. It was higher than those reported for other bioanalytical methods dedicated to the analysis of fosfomycin [19,20,21,23,24,25].

The simple extraction procedure additionally produced clean sample extracts for injection into the chromatographic system. Sample dilution played a crucial role in preventing the carry-over effect between runs and in reducing fouling effects on the chromatographic system.

The dynamic range of concentrations validated for this method was 0.1 μg/g to 20 μg/g and allowed for direct and accurate measurement of fosfomycin concentrations in most prostatic tissue samples, according to the distribution retrieved in patients in the validation cohort. The wide range of values observed (1.2–78.3 μg/g) is consistent with the spread of values for F concentration observed in other biological samples, i.e., plasma and urine [16]. Samples in which concentrations exceeded 20 μg/g were retrieved and reanalyzed after a dilution step with a blank homogenate. This was made possible as a result of the successful dilution integrity experiments, which were performed up to 80 μg/g.

The present method showed a noteworthy sensitivity, even according to the S/N ratio = 88.5 determined at the lowest level (0.1 µg/g) in the dynamic range of calibration. The high S/N ratio value indicates that a higher level of sensitivity could have been reached. However, considering that concentration levels below 0.1 µg/g are not useful for clinical purposes, a further improvement in the sensitivity of the assay was not pursued.

It is noteworthy that the sensitivity of the present assay may allow us to analyze specimens weighing only a few milligrams. This may represent a relevant topic from a clinical point of view, allowing for the determination of F concentrations not only in large tissue samples, such as those collected after TURS and/or Holep procedures, but also in small samples collected after prostatic biopsies. In this validation study, however, the method was only applied to bioptic tissue samples collected with a sparse strategy from patients undergoing TURP or HoLEP, which allowed us to collect data for developing a population PK/PD model for predicting the probability of effective fosfomycin exposure in prostatic tissue after oral F-t administration in clinics.

Careful management of samples is a mandatory indication in good laboratory practices, and it is herein recommended in order to minimize the occurrence of errors in F determination due to the instability of the molecule itself. Indeed, our findings indicate that some critical conditions exist, despite fosfomycin being generally considered a stable molecular entity. Specifically, only one freeze–thaw cycle was safe before a certain decrease in F concentrations was observed. In our practice, we therefore recommend the prompt delivery of samples to the lab in order to avoid unnecessary freezing at −80 °C before analysis, and any consequent occurrence of freeze–thaw cycles in cases of reanalysis.

The limitations of our study should be acknowledged. One is related to the addition of the FIS to the extraction–precipitation solution. On the one hand, this operational choice most probably did not allow for the full equilibration of the internal standard with the biological matrix. On the other hand, true mimicking of F incorporation into the tissue could only be achieved by administering the FIS to the patients, which was clearly not possible in practice. However, the high extraction yield (>90%) observed at all concentration levels was most probably preventing any inaccuracy arising from improper use of the FIS. Another limitation in our present study is clearly related to the fact that no assessment of F concentrations in the different zones of the prostatic tissue was performed.

## 5. Conclusions

In conclusion, in this study we developed a bioanalytical method for measuring fosfomycin concentrations in human prostatic tissue homogenate and fully validated the method according to EMA criteria. The straightforward analytical procedure, coupled to the sensitivity of the LC-MS/MS technique, may allow for the analysis of prostatic samples weighing only a few milligrams, thus potentially ensuring accurate determination of prostatic fosfomycin concentrations for clinical purposes. This novel assay could also be useful for implementing a population pharmacokinetic/pharmacodynamic (PK/PD) model for predicting the probability of target attainment of effective fosfomycin exposure in prostatic tissue after oral F-t administration.

## Figures and Tables

**Figure 1 pharmaceutics-16-00681-f001:**
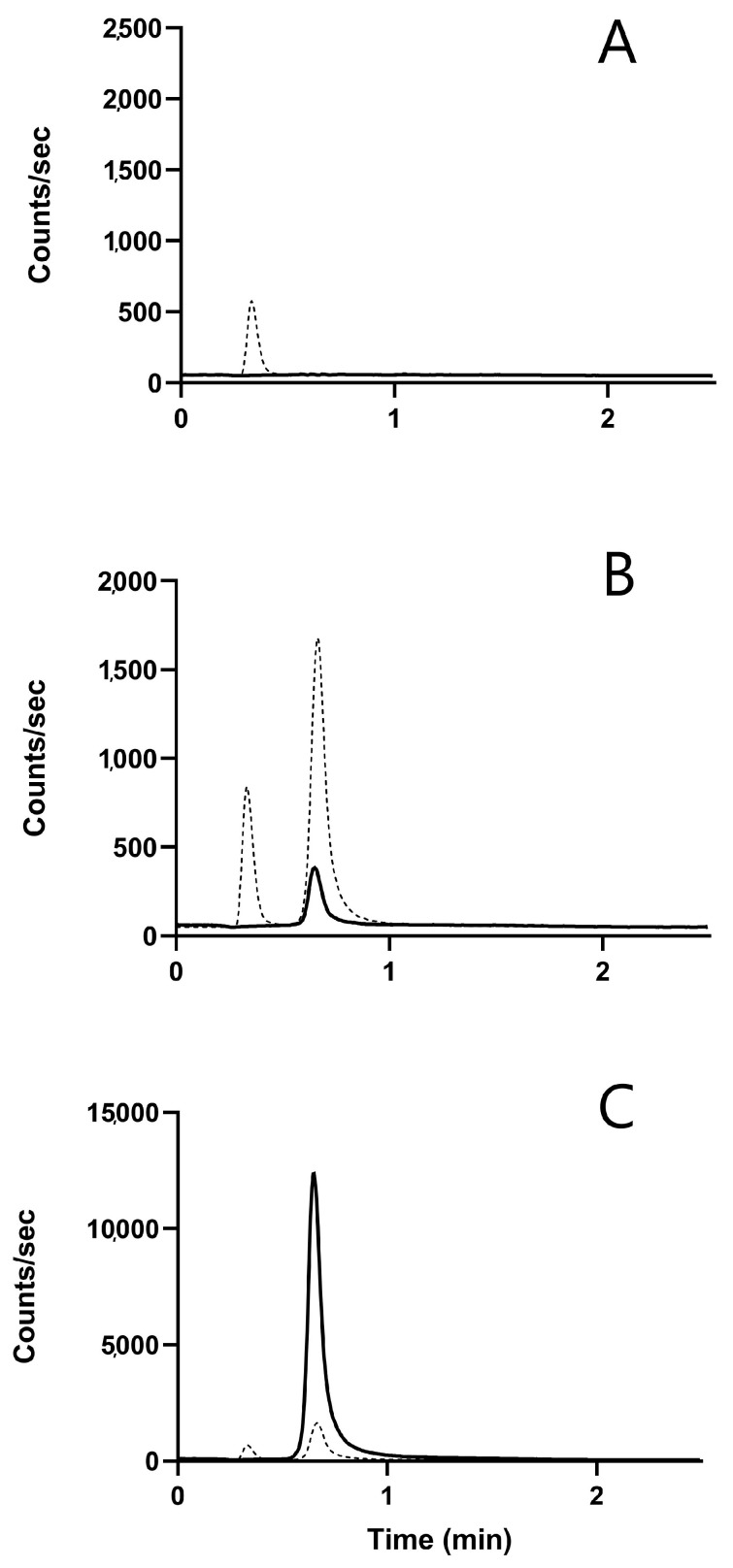
Overlain MRM chromatograms for F (black line) and FIS (dashed line) obtained in the analysis of: (**A**) a blank tissue extract, showing the absence of peaks related to F and the presence of an isobaric peak related to the FIS at 0.4 min; (**B**) an LLOQ sample with an S/N ratio = 88.5 for the F peak at 0.7 min; and (**C**) a real patient sample showing a good peak shape and resolution for the F and IS peaks.

**Figure 2 pharmaceutics-16-00681-f002:**
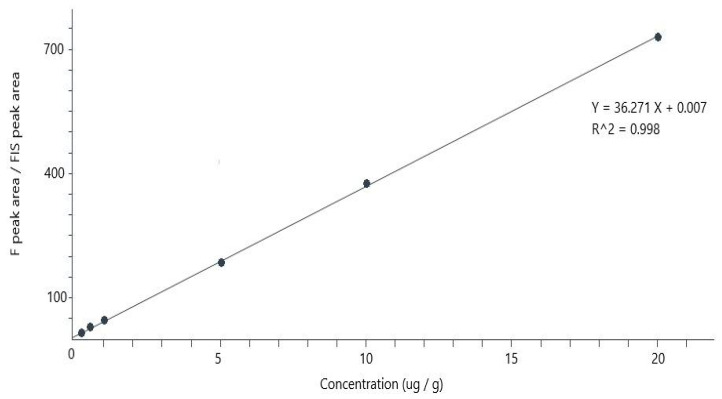
Example of a calibration graph obtained by plotting the F/IS area ratio (response) over concentrations in the 0.1–20.0 µg/g range; and software fitting of 6 experimental calibration points with the linear equation and correlation coefficient reported in the upper-right corner.

**Figure 3 pharmaceutics-16-00681-f003:**
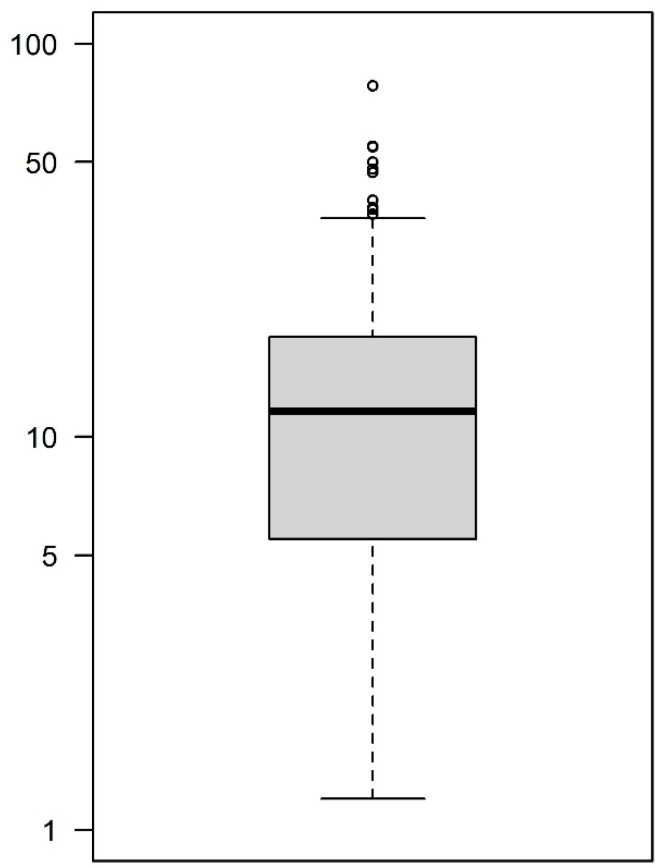
BoxPlotR showing the spread of the F concentration measured in real patients’ prostatic samples. Population size: 105; Median: 11.4 μg/g; Mean: 14.7 μg/g; Minimum: 1.2 μg/g; Maximum: 78.3 μg/g; 25th quartile: 5.4 μg/g; 75th quartile: 17.95 μg/g.

**Table 1 pharmaceutics-16-00681-t001:** Specific MRM transition parameters used for F and IS acquisition.

Analyte	Retention Time (min)	Precursor Ion (*m*/*z*)	Product Ion (*m*/*z*)	Dwell Time (ms)	Fragmentator (eV)	Collision Energy (eV)
F	1.21	137.0	79.0	100	166	33
IS	1.20	140.0	79.1	100	166	33

**Table 2 pharmaceutics-16-00681-t002:** Binary pump program used for linear gradient elution with mobile phases A and B.

Time (min)	A (%)	B (%)	Flow (mL/min)
0.00	95.00	5.00	0.500
1.00	80.00	20.00	0.500
1.50	5.00	95.00	0.500
2.00	5.00	95.00	0.500
2.01	95.00	5.00	0.500
2.50	95.00	5.00	0.500

**Table 3 pharmaceutics-16-00681-t003:** Intra-day and inter-day average (avg) precision and accuracy assessed at three concentration levels (LQC, MQC and HQC) five times (intra-day) in three different analytical runs (inter-day) for F.

		Intraday (n = 5)			Inter-Day (n = 3)		
QC Levels	Nominal Conc. (µg/g)	Avg Conc. (µg/g)	Avg Precision (CV%)	Avg Accuracy (Bias%)	Avg Conc. (µg/g)	Avg Precision (CV%)	Avg Accuracy (Bias%)
LQC	0.25	0.24	8.6	9.1	0.24	9.9	8.8
MQC	2.5	2.6	2.8	9.9	2.5	6.8	1.7
HQC	12.5	12.4	8.2	8.2	12.4	4.1	4.1

**Table 4 pharmaceutics-16-00681-t004:** Average (n = 30) matrix effect (ME%) and recovery (ER%) and IS-normalized ME% and ER%, measured at different concentration levels.

Quality-Control Level	N°	ME (%)	ISn-ME (%)	ER (%)	ISn-ER (%)
LQC	30	65.7	93.4	97.4	100.2
MQC	30	68.5	94.5	96.8	99.5
HQC	30	69.2	95.6	97.8	100.4

**Table 5 pharmaceutics-16-00681-t005:** Stability of F at different storage conditions. In our study, we tested both the extracts and the samples (according to our routine needs).

Quality-Control		LQC	MQC	HQC
Types of sample	Tested conditions	Avg Accuracy (Bias%)	Avg Accuracy (Bias%)	Avg Accuracy (Bias%)
extracts	Autosampler after 12 h	−5.1	−5.5	−5.2
	Freezer (−20 °C) after 24 h	−4.5	−2.7	−3.8
Tissue samples		Freeze-thaw stability (from −80 °C to room temp.)		
	1 cycle	−8.2	−8.6	−8.8
	2 cycle	−12.6	−12.2	−12.5
	3 cycle	−27.1	−25.2	−26.1

## Data Availability

The data presented in this study are available on request from the corresponding author. The data are not publicly available due to privacy concerns.

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
