# Peer review of "A Sensitive Liquid Chromatography–Tandem Mass Spectrometry Method for Measuring Fosfomycin Concentrations in Human Prostatic Tissue"

_pharmaceutics, 2024, doi:10.3390/pharmaceutics16050681_

Round 1

Reviewer 1 Report

Comments and Suggestions for Authors

The article, A Very Sensitive Liquid Chromatography-Tandem Mass Spectrometry Method for Measuring Fosfomycin Concentrations in Human Prostatic Tissue by Conti et al. describes an LC-MS/MS assay for the quantification of Fosfomycin in prostate tissues. The study is more of a short communication that does not contribute anything unique to the literature. There are several issues that need to be addressed to get this limited study into a form that can be published.

Several assays for Fosfomycin have been published. The stated reason for the study is that a very sensitive assay for fosfomycin in prostatic tissue is needed but the study does not seem to support this. The LLOQ is 0.1ug/g but the lowest concertation measured in Figure 3 is 1.2 ug/g more than 10x the LLOQ. This suggests that assays with this kind sensitivity are not needed. The introduction needs to be rewritten with a better justification.  

The assay should be compared to some of the many LC-MS assays that have been developed such as reference 13.

Table 2: Specific MRM transition parameters used for F and IS acquisition

The units for collision energy should be in acceptable units for energy which is typically the electron volt eV and not V as listed in Table 2.

The column “Fragmentator” is not defined. What is this value, is it a declustering potential, entrance potential or a collision cell exit potential?  

2.6.2. Linearity and limit of quantification (LOQ)

With respect to the calculation of the LLOQ, S/N needs to be defined since S/N can be calculated several different ways.   

Results

3.1. Optimization of LC-MS/MS conditions

Precursor and product ion masses for the fosfomycin and the IS are unnecessarily repeated, since the finalized parameters are enumerated in Table 2 and in any case are not represented correctly. They appear as “137.0-79.0 and 140.0-79.0 m/z”, respectively whereas they should be written 137.0 >79.0 and 140.0>79.0 m/z.

Figure 1 should be a single figure not divided over 2 pages. The IS and F peaks need to be clearly delineated either by different colors or making one line dashed, dotted etc. Each chromatogram needs to be labeled A, B,C. For Figure 1B it is not clear to what peak the S/N 88.5 referrers to. Both fosfomycin and IS peaks should have an S/N listed.  Figure 1A had two peaks. The authors need to identify which peak corresponds to the IS. If the peak at 0.3min in the IS channel is in blank tissues, then the authors should show a chromatogram of the tissue extract with no IS.  

3.2.3. Linearity and Dilution Integrity

The linear equation has no intercept which is very unusual, the authors need to justify why the calibration curve was forced through zero. I have a hard time believing that the fit is worse with an intercept.

Was the data for the calibration weighted? Typically, calibration curve data is weighted with 1/x or 1/x^2. If not weighting was done authors need to justify why no weighting was done.

Figure 2: The fronts are too small. There is a vertical line on the left hand side that needs to be removed. The y-axis units are not clear. What is the relationship between relative responses and standard IS peak area ratio? The x-axis should not stop numbering after 16. The units of mcg are non-standard and the Greek letter mu should be used in place of mc. The calibration line should not be extrapolated beyond the highest calibration point.

Table 3. The QC and avg con. are in units of mg/L while the calibration is in units of mass and not volume. Why is this the case?

Figure 3.

For the y-axis label the Greek letter mu should be used in place of mc.

In Figure 3 Several points are above the dilution integrity of 40ug/g. A new dilution integrity of up to 80ug/g and corresponding CV% needs to be performed.

Comments on the Quality of English Language

There are several instances of very confusing English construction. Eg

“For this latter purpose, it to have available a very sensitive method”

Author Response

Please check file attached

Reviewer 2 Report

Comments and Suggestions for Authors

The method proposed in this paper seems simple, fast, and efficient for the quantification of fosfomycin in prostatic samples. I think that more attention could be paid to the relevance of monitoring the antibiotic in the tissue to reinforce the paper. Regarding the sample collection to be minimally invasive, the minimal amount of tissue necessary for the analysis is not given. I guess that the samples come from prostatectomy, so I wonder if the analysis would be viable from samples from biopsies in which the sample amount available is limited. The authors should clarify these points.

Specific comments

Title: "Very" is ambiguous. It can be deleted.

The abstract (too) is concise. Some quantitative outcomes could be added.

Introduction: Several recent methods for the determination of fosfomycin in biological matrices have not been cited. It would be worth commenting on the state of the art of this topic.

Section 2.5. "Multiple Reaction Monitoring"

Section 2.6 and throughout the manuscript: "n" for the number of replicates should be lowercase in italics.

Section 3.1. “… literature [17–22]Optimization”. Add full stop and space.

Figure caption 2 (and throughout the entire paper). Revise the use of commas to indicate decimal figures.

As I have commented above, the discussion of results can be complemented by comparing the performance of other analytical methods for similar purposes.

The conclusions are quite poor. I suggest the authors put a little more effort into enriching this section. You can comment on aspects of pros and cons, for example.

There are some typographical errors in the text, such as postatic in the intro, that should be corrected.

Comments on the Quality of English Language

The language is, in general, good. There are some typos to be corrected.

Author Response

Please see file attached

Round 2

Reviewer 2 Report

Comments and Suggestions for Authors

As far as I'm concerned, the paper can be accepted.